# Exploring the Interplay of RUNX2 and CXCR4 in Melanoma Progression

**DOI:** 10.3390/cells13050408

**Published:** 2024-02-27

**Authors:** Luca Dalle Carbonare, Arianna Minoia, Anna Vareschi, Francesca Cristiana Piritore, Sharazed Zouari, Alberto Gandini, Mirko Meneghel, Rossella Elia, Pamela Lorenzi, Franco Antoniazzi, João Pessoa, Donato Zipeto, Maria Grazia Romanelli, Daniele Guardavaccaro, Maria Teresa Valenti

**Affiliations:** 1Department of Engineering for Innovative Medicine, University of Verona, 37134 Verona, Italy; luca.dallecarbonare@univr.it (L.D.C.); arianna.minoia@univr.it (A.M.); anna.vareschi@univr.it (A.V.); sharazed.zouari@univr.it (S.Z.); 2Department of Neurosciences, Biomedicine and Movement Sciences, University of Verona, 37134 Verona, Italy; francesca.cristiana@univr.it (F.C.P.); mirko.meneghel@univr.it (M.M.); pamela.lorenzi@univr.it (P.L.); donato.zipeto@univr.it (D.Z.); mariagrazia.romanelli@univr.it (M.G.R.); 3Department of Surgery, Dentistry, Paediatrics and Gynaecology, University of Verona, 37134 Verona, Italy; alberto.gandini@univr.it (A.G.); franco.antoniazzi@univr.it (F.A.); 4Department of Medicine, University of Verona, 37134 Verona, Italy; rossella.elia@univr.it; 5Department of Medical Sciences and Institute of Biomedicine—iBiMED, University of Aveiro, 3810-193 Aveiro, Portugal; joao.pessoa@ua.pt; 6Department of Biotechnology, University of Verona, 37134 Verona, Italy; daniele.guardavaccaro@univr.it

**Keywords:** melanoma, RUNX2, CXCR4, transcription factor

## Abstract

Overexpression of the Runt-related transcription factor 2 (*RUNX2*) has been reported in several cancer types, and the C-X-C motif chemokine receptor 4 (*CXCR4*) has an important role in tumour progression. However, the interplay between *CXCR4* and *RUNX2* in melanoma cells remains poorly understood. In the present study, we used melanoma cells and a *RUNX2* knockout (*RUNX2*-KO) in vitro model to assess the influence of *RUNX2* on CXCR4 protein levels along with its effects on markers associated with cell invasion and autophagy. Osteotropism was assessed using a 3D microfluidic model. Moreover, we assessed the impact of CXCR4 on the cellular levels of key cellular signalling proteins involved in autophagy. We observed that melanoma cells express both *RUNX2* and *CXCR4*. Restored *RUNX2* expression in *RUNX2* KO cells increased the expression levels of *CXCR4* and proteins associated with the metastatic process. The protein markers of autophagy LC3 and beclin were upregulated in response to increased *CXCR4* levels. The CXCR4 inhibitor WZ811 reduced osteotropism and activated the mTOR and p70-S6 cell signalling proteins. Our data indicate that the RUNX2 transcription factor promotes the expression of the CXCR4 chemokine receptor on melanoma cells, which in turn promotes autophagy, cell invasiveness, and osteotropism, through the inhibition of the mTOR signalling pathway. Our data suggest that *RUNX2* promotes melanoma progression by upregulating *CXCR4*, and we identify the latter as a key player in melanoma-related osteotropism.

## 1. Introduction

The treatment of melanoma, an aggressive form of skin cancer, is a challenge for the medical and scientific communities. In recent years, extensive research has focused on identifying key molecules and biochemical mechanisms that contribute to melanoma progression. In this framework, the Runt-related transcription factor 2 (RUNX2) and the C-X-C motif chemokine receptor 4 (CXCR4) have emerged as proteins of interest, being associated with the migration, invasion, and metastasis of tumour cells.

RUNX2, a transcription factor related to osteogenesis, has been identified as an important protein involved in melanoma progression [1,2,3,4]. Recent studies have shown that *RUNX2* expression was upregulated in advanced melanomas and correlated with poor patient prognosis [5,6]. RUNX2 promotes cell adhesion, invasion, and metastasis through the activation of target genes involved in extracellular matrix degradation. Furthermore, it has been demonstrated that RUNX2 could interact with other key proteins, such as p53 and Wnt/β-catenin, further amplifying the protumoral effects in melanoma [7,8].

In addition, CXCR4, a chemokine receptor, has been shown to play a significant role in the invasion and metastasis properties of melanoma [9,10]. *CXCR4* is expressed on melanoma tumour cells. Its ligand, the CXCL12 chemokine, is present in many organs identified as common sites of melanoma metastasis [11]. The interaction between CXCR4 and CXCL12 promotes the migration of tumour cells, their invasion into surrounding tissues, and the formation of new metastases [12].

Thus, RUNX2 and CXCR4 represent two crucial molecules involved in melanoma progression. Experimental evidence has demonstrated their role in promoting the migration, invasion, and metastasis of tumour cells [13]. Therefore, the inhibition of CXCR4, or its cell signalling function, is a potential therapeutic strategy against melanoma. Gaining a deep understanding of the underlying mechanisms of these interactions may lead to the development of new targeted therapeutic strategies that could improve the prognosis of melanoma patients.

In this study, we aimed to investigate the role of RUNX2 and CXCR4 in melanoma, using in vitro cell cultures. We modulated the levels of RUNX2 in melanoma cells and assessed its effects on CXCR4 as well as on markers of cell invasion and autophagy. We also assessed its impact on osteotropism and on the mTOR signalling pathway. Since these proteins and processes are involved in tumour progression and metastasis, our findings contribute critical insight into the pathophysiology of melanoma.

## 2. Material and Methods

### 2.1. Cell Cultures

The malignant melanoma cell lines A375, Colo 800, Colo 853, Colo 679, and MELHO used in this study were obtained from American Type Culture Collection (ATCC; Manassas, VA, USA) or from Sigma-Aldrich (St. Louis, MO, USA). The *RUNX2* knockout (KO) cell lines were generated using CRISPR/Cas9, as previously described [4]. These KO cell lines were named 1E7, 1B3, and 1F5. Cell lines were cultured in RPMI 1640 growth medium (Sigma-Aldrich, St. Louis, MO, USA) supplemented with 10% foetal bovine serum (FBS; Sigma-Aldrich, St. Louis, MO, USA), 1% penicillin/streptomycin antibiotics, and 1% glutamine. The primary human osteoblast cell line (HOB) was purchased from Promocell (Heidelberg, Germany; reference: C-12720) and cultured with osteoblast growth medium (PromoCell, Heidelberg, Germany; reference: C-27001).

The cells were maintained in a 5% CO_2_ humidified incubator at 37 °C. To ensure the absence of mycoplasma contamination, all cell lines were tested using the LookOut Mycoplasma PCR Detection Kit (Sigma-Aldrich, St. Louis, MO, USA), following the manufacturer’s instructions. Negative results were obtained for all cell lines, confirming their mycoplasma-free status.

The CXCR4 inhibitor WZ811 (Selleck, Houston, TX, USA) was used at a concentration of 10 μM, determined through an XTT test (Appendix A), according to the manufacturer’s instructions. In particular, we used the Cell Proliferation Kit II (XTT, Invitrogen, Waltham, MA, USA), as previously described [14]. We chose the concentration of 10 µM, as cellular viability remained unaltered.

To assay the autophagy markers, we treated the cells with 10 nM bafilomycin A1 (Sigma-Aldrich, St. Louis, MO, USA) or 1 mM 3-methyladenine (Sigma, Shanghai, China) for 6 h, as previously reported [15,16].

### 2.2. RUNX2 Cloning and Transfection

The *RUNX2* gene was cloned into the pcDNA3 expression vector, as we previously reported [4]. The amplified *RUNX2* gene sequence was initially inserted into the pCRTM2.1 cloning vector (Thermo Fischer Scientific, Waltham, MA, USA), then excised using EcoRV/XhoI digestion and cloned into the pcDNA3-Flag-HA vector (Addgene plasmid #10792, Watertown, MA, USA) [4]. Transfection of the RUNX2-expressing plasmid was carried out using RUNX2 Lenti ORF particles from Origene Technologies (Rockville, MD, USA), following the manufacturer’s instructions. *RUNX2* KO cells were grown to 70% confluence and subsequently transfected with 1.3 µg/mL or 2.6 µg/mL RUNX2-expressing pcDNA3 vector, using the Lipofectamine 3000 reagent (Thermo Fisher Scientific, Waltham, MA, USA; reference: L3000-008), as we previously reported [17].

### 2.3. Immunofluorescence

Immunofluorescence analyses were performed as we previously reported [4]. Briefly, cells were fixed with 4% of PFA, permeabilised with Triton-X 0,5% and processed according to the manufacturer’s protocols. The primary antibody RUNX2 -Sant Cruz-8566 (Santa Cruz Biotechnology, Inc., 10410 Finnell Street Dallas, Dallas, TX, USA)was diluted 1:100 (as reported in the datasheet) in antibody dilution buffer (BSA) and incubated overnight at 4 °C. Slides were then incubated with secondary antibodies anti-goat in FITC-Sant Cruz-2024 (Santa Cruz Biotechnology, Inc., 10410 Finnell Street Dallas, Dallas, TX, USA) diluted 1:300 (as reported in the datasheet). Nuclear staining was performed by ProLong™ Gold Antifade Mountant with DAPI. Images were recorded using a Leica (Wetzlar, Germany) inverted microscope at 10 × 2.3.

### 2.4. Three-Dimensional (3D) Cell Cultures

To investigate the interaction between osteoblasts (HOB) and melanoma cells, 3D cultures were performed using appropriate scaffolds (VITVO^®^, Rigenerand Srl, Medolla, Italy). Before inserting the cells into the VITVO^®^ scaffold, HOBs and melanoma cells were separately cultured for three days in flasks containing osteogenic or complete RPMI medium, respectively. On the fourth day, the cells were stained with vital fluorescent dyes (Vybrant Cell Labeling Solution, Invitrogen, Waltham, MA, USA). HOB cells were stained with 1,1′-dioctadecyl-3,3,3′,3′-tetramethylindocarbocyanine perchlorate (DiI, emission at 565 nm, red), while melanoma cells were stained with 3,3′-dioctadecyloxacarbocyanine perchlorate (DiO, emission at 501 nm, green). Then, HOB cells were injected into the VITVO^®^ scaffolds. The 3D fluidic model was performed as follows: HOB cells were cultured within the device, which was connected to the peristaltic pump, to form a closed-loop fluidic circuit. The circuit contained 3.8 mL medium with melanoma cells in circulation for 1 h, at a rate of 0.3 cm/s, to simulate the capillary flow rate. The scaffolds were observed using an EVOS fluorescence microscope (Life Technologies, Carlsbad, CA, USA). Images were captured using a green fluorescent protein filter to visualise HOB cells and a red fluorescent protein filter to visualise melanoma cells. Merged pictures with both filters activated were also taken. To evaluate the number of melanoma cells (red) infiltrated into HOB cells (green), six randomly selected squares with 100^2^ µm dimensions were analysed using Image J software (Version 1.46r, National Institutes of Health, Bethesda, MD, USA).

### 2.5. Total RNA Extraction and Reverse Transcription (RT)

Total RNA extraction and reverse transcription (RT) were performed using previously reported methods [17]. Briefly, cells were harvested and total RNA was isolated using the RNeasy^®^ protect mini kit (Qiagen, Hilden, Germany), according to the manufacturer’s instructions. The concentration and quality of RNA were determined using the Qubit™ RNA HS assay kit’ (Invitrogen, Waltham, MA, USA) and a Qubit 3 fluorometer (Thermo Fisher Scientific, Waltham, MA, USA; reference: Q3321). Reverse transcription (RT) was carried out using the first-strand cDNA synthesis kit (GE Healthcare, Chicago, IL, USA) to convert the extracted RNA into complementary DNA (cDNA). This cDNA served as the template for subsequent real-time quantitative PCR analysis.

### 2.6. Real-Time Quantitative PCR

Real-time quantitative PCR (RT-qPCR) was performed to evaluate gene expression levels. PCR reactions were conducted in a total volume of 25 µL, consisting of 2.5 µL of cDNA, PCR master mix with carboxyl-X-rhodamine (ROX) and the predesigned, gene-specific primers and probe sets obtained from assay-on-demand gene expression products (Thermo Fisher Scientific, Waltham, MA, USA) and targeting the following genes of interest: *RUNX2* (Hs1047973_m1), *SPP1* (hs00167093_m1), *CXCR4* (Hs00976734_m1), *MMP3* (Hs00968305_m1), and *B2M* (hs999999_m1 [housekeeping gene control]).

The thermal cycling conditions included an initial denaturation step followed by 40 amplification cycles, as previously reported [17]. The fluorescence signal was captured and analysed using a real-time PCR system. The expression levels of the target genes were normalised to appropriate reference genes, and relative gene expression levels were determined using the comparative cycle threshold (Ct) method: relative RNA level = 2^−(ΔCt target − ΔCt reference)^.

### 2.7. Western Blotting

Protein extraction was performed in radioimmunoprecipitation (RIPA) buffer (Thermo Fisher Scientific, Waltham, MA, USA), according to previously reported methods [18]. The protein concentrations were determined using a BCA assay kit (Thermo Scientific, Waltham, MA, USA). Briefly, protein samples were diluted in Laemmli’s sample buffer (Bio-Rad, Hercules, CA, USA) and heated at 95 °C for 5 min. The proteins were then separated using 4–20% SDS-PAGE Mini-PROTEAN TGX Precast Gels-(Bio-Rad, Hercules, CA, USA). After electrophoresis, the proteins were transferred onto polyvinylidene difluoride (PVDF) membranes (Thermo Fisher Scientific, Waltham, MA, USA).

The PVDF membranes were probed with specific primary antibodies, to detect RUNX2 (1:1000 dilution; Cell Signaling Technology, Danvers, MA, USA, reference: 8486), CXCR4 (1:100 dilution; Abcam, Cambridge, UK, reference: ab124824), MMP13 (1:500 dilution; GeneTex, Irvine, CA, USA, reference: GTX 100665), RANKL (1:500 dilution; Santa Cruz Biotechnology, Dallas, TX, USA, reference: sc377079), LC3B (1:1000 dilution; Invitrogen, Waltham, MA, USA, reference: PA5-22939), p62/SQSTM1 (1:1000 dilution; Rockland, Baltimore, MD, USA, reference: 600-401-HB8), beclin 1 (1:1000 dilution; GeneTex, Irvine, CA, USA, reference: GTX133555), mTOR (1:1000 dilution; Cell Signaling Technology, Danvers, MA, USA, reference: 2983), P-mTOR (1:1000 dilution; Cell Signaling Technology, Danvers, MA, USA, reference: 5536), p70-S6 (1:1000 dilution; Cell Signaling Technology, Danvers, MA, USA, reference: 9202), P-p70-S6 (1:500 dilution; Cell Signaling Technology, Danvers, MA, USA, reference: 9205), and β-actin (1:10,000 dilution; Thermo Scientific, Waltham, MA, USA, reference: BA3R). Following primary antibody incubation, membranes were further incubated with appropriate secondary antibodies, such as anti-mouse (1:2000 dilution; Cell Signaling Technology, Danvers, MA, USA, reference: 7076) or anti-rabbit (1:2000 dilution; Cell Signaling Technology, Danvers, MA, USA, reference: 7074) horseradish peroxidase-conjugated antibodies.

An enhanced chemiluminescence reagent (ECL; Millipore, Burlington, MA, USA) was used to detect the antibody–protein complexes. Images were captured using a LAS4000 Digital Image Scanning System (GE Healthcare, Little Chalfont, UK). Densitometric analysis of the bands was performed as previously described [18].

### 2.8. Statistical Analysis

Statistical analysis was performed using SPSS for Windows, version 22.0 (SPSS Inc.; Chicago, IL, USA). The data obtained from the experiments were expressed as the mean ± standard deviation (SD). Statistical analysis was performed using a two-tailed Student’s paired *t*-test to estimate the significance of the differences between controls and the respective experimental conditions. A *p*-value of less than 0.05 was considered statistically significant. For in vitro experiments, data analysis was conducted based on at least three independent assays.

## 3. Results

### 3.1. CXCR4 Is Associated with RUNX2 Expression, Melanoma Invasiveness, and Osteotropism

To evaluate RUNX2 association with CXCR4 in melanoma cells, we started by detecting the expression of these two markers and the matrix metalloproteinase 13 (MMP13) invasiveness marker in several melanoma cell lines. The Colo 800, Colo 853, Colo 679, A375, and MELHO cell lines expressed *RUNX2* (Figure 1a), the *MMP13* invasiveness marker (Figure 1b), and *CXCR4* (Figure 1c), at the messenger RNA (mRNA) level. The expression of *RUNX2*, *MMP13* and *CXCR4* at the protein level was also verified in the same melanoma cell lines (Figure 1d,e). These observations confirm the expression of these three proteins as a conserved feature of melanoma cells.

To evaluate the effect of the RUNX2 transcription factor on *CXCR4* expression, we used three *RUNX2* knockout (KO) MELHO cell lines generated by CRISPR-Cas9 gene editing: 1E7, 1B3, and 1F5. Western blotting analysis confirmed that the RUNX2 protein was not expressed in any of the three KO cell lines where the *RUNX2* gene was deleted (Figure 2a). Immunofluorescence further confirmed this observation in the three *RUNX2* KO cell lines (Figure 2b). We further analysed the *CXCR4* expression levels in the three *RUNX2* KO cell lines. Strikingly, real-time quantitative PCR (Figure 2c), as well as Western blotting analysis (Figure 2d), revealed that *CXCR4* was downregulated in all of the three *RUNX2* knockout cell lines. This observation indicated that *RUNX2* is involved in *CXCR4* expression.

We further assessed the association between the expression levels of *RUNX2* and *CXCR4* through RUNX2 overexpression. To perform functional assays, we used the IF5 *RUNX2* KO cell line. We restored RUNX2 protein levels in this cell line through the ectopic expression of RUNX2 from a transiently transfected pcDNA3/RUNX2 plasmid, using two different plasmid concentrations: 1.3 µg/mL and 2.6 µg/mL.

Real-time quantitative PCR analyses showed higher *CXCR4* expression levels in 1F5 *RUNX2* KO cells transfected with both plasmid concentrations (Figure 3a). The mRNA expression levels of the secreted phosphoprotein 1 (SSP1) serine/threonine protein kinase, a transcriptionally regulated *RUNX2* gene [19] and an additional invasiveness marker, as well of matrix metalloproteinase 3 (*MMP3*), were also increased in *RUNX2*-restored 1F5 cells (Figure 3a).

In addition, the protein markers associated with cell invasiveness, receptor activator of nuclear factor κΒ ligand (RANKL; Figure 3b) and MMP13 (Figure 3c), as well as CXCR4 (Figure 3c), were expressed at higher levels in RUNX2-restored 1F5 cells. These observations indicate that *CXCR4* expression is promoted by RUNX2 and is associated with increased cell invasiveness properties.

After demonstrating the effect of RUNX2 on the expression levels of *CXCR4*, we sought to assess CXCR4’s impact on the formation of melanoma metastasis in the bone. To evaluate the role of CXCR4–RUNX2-dependent osteotropism in melanoma cells, we used a 3D cell culture model. The system was connected to a compartmentalised fluidic device, where we evaluated the percentage of melanoma cells adherent to osteoblast cells. Specifically, 24-well cell culture inserts containing human osteoblast cells (HOBs) were placed and cultured within the device, connected to a peristaltic pump, to form a closed-loop fluid circuit. The fluid circuit contained 3.8 mL of circulating medium with melanoma cells (MELHO, 1F5, or 1F5/RUNX2++ with or without WZ811), at a velocity of 0.3 cm/s, to simulate capillary flow. As shown in Figure 4a (top), the percentage of 1F5 *RUNX2* KO cells adhering to HOBs was lower than that of wild-type MELHO cells. When RUNX2 expression was restored in *RUNX2* KO cells (RUNX2++1F5), we observed higher cell adhesion to HOBs than in the 1F5 *RUNX2* KO cell line. However, this enhanced osteotropism effect was cancelled in the presence of the CXCR4 inhibitor WZ811 (Figure 4a; down). In this system, the quantification of melanoma cell adherence to osteoblasts confirms their decreased infiltration level in the absence of RUNX2 or under CXCR4 inhibition (Figure 4b). These effects were correlated with the mRNA levels of *MMP3* in the cells (Figure 4c). MMP3 is a protein involved in cancer-induced bone lesions [20]. The inhibition of CXCR4 with WZ811 lowered *MMP3* mRNA expression to levels comparable to those obtained in the absence of the *RUNX2* gene (Figure 4c). These observations indicate that *RUNX2* expression in melanoma cells is essential for their adherence to bone cells, suggesting an enhanced capacity for cell invasion and bone metastasis formation. Importantly, the CXCR4 protein levels in RUNX2-restored 1F5 cells were comparable with those from their parental MELHO cell line (Figure 4d).

### 3.2. RUNX2 Expression Upregulates Autophagy Markers in Melanoma Cells

Bone metastasis has been associated with increased levels of autophagy [21]. Thus, we tested if the expression of *RUNX2* in melanoma cells could modulate this process. First, we compared the levels of two autophagosome markers, microtubule-associated protein 1A/1B-light chain 3 (LC3) and p62, on the MELHO wild-type melanoma cell line and in the three *RUNX2* KO cell lines: 1E7, 1B3 and 1F5. In comparison with MELHO, the levels of LC3 were decreased in the three *RUNX2* KO cell lines. On the other hand, the levels of p62 followed less consistent variations (Figure 5a). The observation of LC3 suggests decreased autophagy in the absence of *RUNX2* expression.

Then, we monitored the protein levels of LC3 and p62 in the presence of the V-ATPase inhibitors bafilomycin A1 or 3-methyladenine. V-ATPases acidify lysosomes, promoting autophagosome degradation. To quantify protein levels, Western blot data were analysed by densitometry. In comparison with untreated cells, the V-ATPase inhibitor bafilomycin A1 induced an increased accumulation of LC3 in both MELHO and 1F5 cells (Figure 5b). On the other hand, the effects of 3-methyladenine were not fully consistent and the alterations of p62 were not statistically significant. The observations on LC3 suggest a higher autophagic flux in parental melanoma cells MELHO.

To further validate the involvement of RUNX2 in autophagy in melanoma cells, we restored RUNX2 protein levels by transfecting 1F5 *RUNX2* KO cells with the pcDNA3/RUNX2 plasmid. In particular, the effects of two different concentrations of transfected pcDNA3/RUNX2 plasmid were investigated. Protein levels were quantified by densitometry analysis of Western blots. We observed that restored *RUNX2* expression in 1F5 cells induced increased protein levels of the autophagy marker LC3 II (Figure 5c). In addition, the autophagy marker beclin was higher in the parental MELHO as well in RUNX2-restored 1F5 cells compared to the 1F5 *RUNX2* KO cells (Figure 5d). These results indicate that *RUNX2* expression is associated with increased levels of autophagy markers and suggest that RUNX2 promotes autophagy in melanoma cells.

### 3.3. CXCR4 Is a Key Player in the RUNX2-Mediated Inhibition of the mTOR Signalling Pathway

After demonstrating that *RUNX2* expression upregulates autophagy markers, we probed its potential regulatory mechanism. The mammalian target of rapamycin (mTOR) signalling pathway is known to inhibit autophagy [22]. As such, using MELHO and 1F5 *RUNX2* KO cells, we hypothesised that the effects of CXCR4 on autophagy could be mediated by the inhibition of this signalling pathway. First of all, we tested whether the inhibition of CXCR4 by WZ811 modulates the pathway that inhibits autophagy in parental MELHO cells. As shown in Figure 6a, we observed that the inhibition of CXCR4 is capable of increasing the levels of phosphorylated p70-S6, which is a mTOR downstream target [23]. Consistent with these data, we observed an increase in phosphorylated mTOR, which inhibits autophagy, when CXCR4 was inhibited in the presence of WZ811 (Figure 6b).

These data thus demonstrate that CXCR4 promotes autophagy in melanoma cells by inhibiting the mTOR/P-mTOR/p70-S6 signalling pathway. To verify the role of RUNX2 in promoting autophagy through CXCR4, we used *RUNX2* KO melanoma cells.

Under RUNX2 overexpression (RUNX2 ++), we observed reduced levels of phosphorylated p70-S6 (P-p70-S6), while the total levels of p70-S6 were unaltered (Figure 7a). This observation suggests that the autophagy induction by RUNX2 involves the inhibition of the mTOR/P-mTOR/p70-S6 signalling pathway. We then assessed the role of CXCR4 in relation to RUNX2 on the mTOR signalling pathway. The inhibition of CXCR4 by WZ811 in RUNX2-overexpressing cells (RUNX2 ++) increased the levels of the P-mTOR and P-p70-S6K phosphorylated kinases to levels comparable to those obtained without RUNX2 expression (Figure 7b).

These increases indicate that CXCR4 inhibition promotes the upregulation of the mTOR signalling pathway. Therefore, these findings demonstrate that the CXCR4 cytokine receptor plays an inhibitory effect on the mTOR signalling pathway.

## 4. Discussion

Melanoma is an aggressive form of skin cancer that can metastasize in distant organs, including the bone. Understanding the molecular mechanisms underlying melanoma invasiveness and its ability to spread to the bone is crucial for developing effective therapeutic strategies.

In this study, we identified that *CXCR4* expression is dependent on *RUNX2* in melanoma cells, shedding light on its role in melanoma invasiveness and osteotropism.

All melanoma cells studied here expressed both *RUNX2* and *CXCR4* at the mRNA and protein levels. We observed that CXCR4 protein levels were significantly reduced in *RUNX2* KO cells, suggesting a strong association between these two markers. It has been demonstrated that the RUNX2 transcription factor upregulates the expression of the chemokine receptor CXCR4, promoting the metastatic ability of human gastric cancer [13]. Accordingly, we observed that the protein levels of the two markers positively correlated with increased cell invasiveness, as demonstrated by the increased levels of MMP13 and RANKL in the RUNX2-restored 1F5 cells. Gene expression analysis revealed an increase in *SSP1*, *CXCR4*, and *MMP3* expression in the RUNX2-restored cells as well, further supporting the association between *CXCR4* and *RUNX2* in promoting invasiveness. By using a microfluidic 3D cell culture model, we demonstrated the role of CXCR4-RUNX2-dependent osteotropism in melanoma cells. In fact, we observed that the RUNX2-restored 1F5 cells exhibited higher adhesion to HOBs, while the addition of the CXCR4 inhibitor WZ811 reduced this osteotropism ability. Moreover, the inhibition of CXCR4 by WZ811 resulted in decreased mRNA expression levels of MMP3, which is a protein involved in cancer-induced bone lesions. In addition to its association with melanoma invasiveness and osteotropism, we also explored the involvement of RUNX2 in autophagy, which is a cellular process associated with bone metastasis [24]. Our findings demonstrated that *RUNX*2 expression increased the protein levels of the LC3 and beclin autophagy markers in melanoma cells.

Autophagy plays a multifaceted role in cancer, acting both as a tumour suppressor and as a promoter of tumour growth and resistance. The context-dependent nature of autophagy in cancer underscores the need for a comprehensive understanding of its regulation and functional implications in different tumour types [25]. Dysregulated autophagy can lead to genomic instability, impaired cell death, and increased susceptibility to oncogenic events [26,27]. In particular, autophagy can promote tumour growth and survival by providing nutrients during metabolic stress and facilitating resistance to therapeutic interventions [28]. Tumour cells exploit autophagy as a survival mechanism to adapt to unfavourable microenvironments, such as nutrient deprivation and hypoxia [29]. This strategy enables cancer cells to maintain high energy levels, mitigate oxidative stress, and evade cell death pathways, ultimately contributing to tumour progression and therapy resistance [30,31].

In melanoma, a complex role of autophagy has been reported [32]. Rosenfeldt et al. demonstrated that reduced autophagy affects melanoma growth in a manner dependent on the phosphatase and tensin homologue (PTEN) status [31]. In contrast, Xie et al. showed that the deletion of autophagy-related 7 (*Atg7*) promotes the growth of melanoma cells [33]. Several studies conducted on melanoma tumours showed higher LC3 levels in metastases compared to primary tumours [34,35]. Our in vitro data suggest an increase in basal autophagy in melanoma cells expressing *RUNX2*. We previously reported that RUNX2 was involved in melanoma metastasis and cell migration [2]. Thus, the findings of increased autophagy in *RUNX2*-expressing melanoma cells suggest the association between autophagy and metastatic ability in melanoma.

Furthermore, to explore the potential role of CXCR4 in RUNX2-induced autophagy, we investigated the mTOR cell signalling pathway. We observed that in comparison with MELHO wild-type cells, *RUNX2* KO cells exhibited higher expression levels of both mTOR and its phosphorylated form, P-mTOR. Restoring *RUNX2* expression in 1F5 *RUNX2* KO cells led to reduced levels of phosphorylated p70-S6K, which is a downstream target of mTOR. This observation suggests that RUNX2 overcomes the autophagy inhibition by modulating the mTOR signalling pathway. However, the inhibition of CXCR4 by WZ811 in the RUNX2-restored cells increased the levels of phosphorylated mTOR and phosphorylated p70-S6K to levels comparable to those observed without *RUNX2* expression, likely inhibiting autophagy.

Our findings shed light on the complex mechanism of RUNX2 in autophagy regulation in melanoma cells. In fact, our data indicate that RUNX2 induces the expression of *CXCR4*, which in turn promotes autophagy, cell invasiveness, and osteotropism, by inhibiting the mTOR signalling pathway in melanoma cells (Figure 8). In the present work, we succeeded in demonstrating the involvement of RUNX2 and CXCR4 in multiple stages of the pathophysiology of melanoma progression and metastasis. These stages include the RUNX2 effect on *CXCR4* mRNA and protein levels (which were upregulated by RUNX2), its impact on pathological properties of melanoma cells (RUNX2 promoted invasiveness, osteotropism, and autophagy), and the associated regulatory mechanism (which involved the inhibition of the mTOR signalling pathway).

Importantly, it has been reported that a CXCR4 ligand, CXCL12, together with the CXCR4 receptor, plays a key role in cancer progression and metastasis [36,37,38]. CXCL12 is ubiquitously expressed in many tissues, and a distribution of CXCL12 similar to that of CXCR4 has been observed in melanoma cells [39]. This distribution may trigger the activation of a CXCL12/CXCR4 axis [40]. Unfortunately, we did not measure the levels of CXCR4 at the membrane level. However, we hypothesise that the modulation of gene expression observed corresponds to a modulation of CXCR4 at the membrane level as well. It is important to note that the CXCL12/CXCR4 axis activates the AKT and ERK cell signalling pathways [41], and we have recently demonstrated that a reciprocal activation between the RUNX2 and AKT/ERK pathways occurs in melanoma [4]. Therefore, the association we observed between CXCR4 and RUNX2 may be an integral part of the CXCL12/CXCR4 cycle. In other words, a high expression of *RUNX2* may increase *CXCR4* levels, which in turn activates the CXCL12/CXCR4 cycle and thus the AKT and ERK signalling pathways. In our *RUNX2* KO cell line (1F5), the resulting decrease in CXCR4 would reduce the autocrine CXCL12/CXCR4 cycle and, consequently, the activation of AKT and ERK. This alteration could explain the lower levels of osteotropism and invasiveness markers we observed in the *RUNX2* KO model, as they are associated with the AKT and ERK signalling pathways. Furthermore, it has been demonstrated that the CXCL12/CXCR4 axis promotes autophagy through the MiR-125b microRNA, which thus functions as an important downstream mediator upon the activation of the CXCL12/CXCR4 axis [42]. In addition, CXCL12 plays a significant role in the induction of autophagy in chondrocytes through the activation of the CXCR4/mTOR signalling axis [43]. In particular, accordingly to our data, the authors demonstrated that CXCR4 induces autophagy by reducing the phosphorylation level of mTOR [43].

Limitations of our work include the lack of additional models, such as human or animal-based models, in order to provide a more comprehensive and robust understanding of the hypothesis’s relevance. Despite these limitations, our study has uncovered new findings for the first time, introducing a new area of interest in melanoma research. Our data represent a pioneering exploration that has revealed previously unknown insights, potentially opening up new ways to understand the mechanisms behind melanoma. Possible future steps could be the quantification of CXCR4 and markers of cell invasiveness, autophagy, and mTOR signalling, in biopsies from metastatic and non-metastatic melanoma patients, by immunohistochemistry. This approach would allow validating our findings in human patients.

Moreover, animal studies could be pursued, in order to test if the transplantation of *RUNX2* KO melanoma cells into mice would lower the rates of tumour progression and metastasis, in comparison with the transplantation of homologous wild-type cells into identical models. These two approaches would provide proof-of-principle data strengthening the value of *RUNX2* and *CXCR4* as potential therapeutic targets against melanoma. Specifically, their downregulation might decelerate disease progression and metastasis in human patients.

In summary, this study highlights the association of *CXCR4* with *RUNX2* expression in melanoma cells and its implications in melanoma invasiveness, osteotropism, and autophagy. These findings provide valuable insights into the molecular mechanisms underlying melanoma progression and association with bone metastasis, which represent a severe pathology, with limited definitive treatment options available. Consequently, there is a critical need to understand the underlying molecular mechanisms to identify potential therapeutic targets. Our data strongly indicate that CXCR4 plays a significant role in the transformation of melanoma cells and suggest that targeting the CXCR4–RUNX2 axis and autophagy pathways may offer potential therapeutic avenues for counteracting melanoma bone metastasis. This finding substantiates a robust foundation for the development of pharmacological interventions tailored for metastatic melanoma with a specific focus on targeting CXCR4. Additionally, this insight not only holds the promise of revolutionising the approach to treating metastatic melanoma but also paves the way for the design and implementation of innovative treatments to address the formidable challenges posed by this condition. Further research is warranted to fully elucidate the complex interplay between CXCR4, RUNX2, and autophagy in melanoma and develop targeted therapies to improve patient outcomes.

## 5. Conclusions

In conclusion, our study highlights the significant role of CXCR4 in driving the transformation of melanoma cells. Targeting the CXCR4–RUNX2 axis and autophagy pathways could offer promising therapeutic strategies against melanoma bone metastasis. This discovery lays a robust foundation for the development of tailored pharmacological interventions, particularly focusing on CXCR4 targeting, to address the challenges of metastatic melanoma. Moreover, this insight not only promises to revolutionise treatment approaches but also opens avenues for innovative therapies to tackle this condition effectively

## Figures and Tables

**Figure 1 cells-13-00408-f001:**
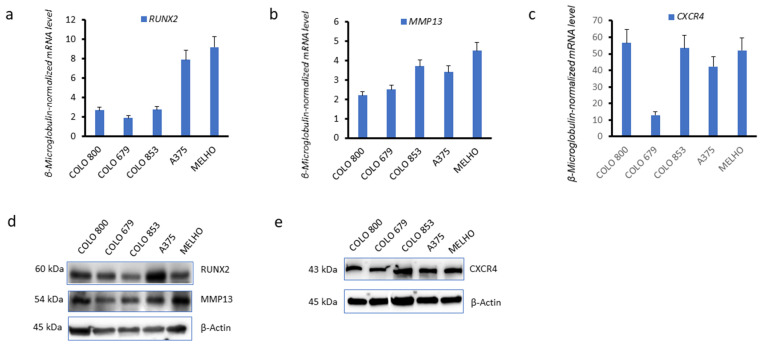
*RUNX2, CXCR4*, and *MMP13* are expressed in melanoma cell lines. (**a**–**c**) Real-time quantitative PCR analysis of messenger RNA (mRNA) expression levels of (**a**) Runt-related transcription factor 2 (*RUNX2*), (**b**) matrix metalloproteinase 13 (*MMP13*) and (**c**) C-X-C motif chemokine receptor 4 (*CXCR4*), in Colo 800, Colo 853, Colo 679, A375, and MELHO cell lines. Data were normalised to the β2-microglobulin housekeeping gene. (**d**,**e**) Western blots of the protein levels of (**d**) RUNX2 and MMP13 and (**e**) CXCR4 in the melanoma cell lines. β-Actin was used as a loading control. mRNA expression levels were evaluated in three independent experiments. Western blot images are representative of three independent experiments. Original blots are presented in Appendix A.

**Figure 2 cells-13-00408-f002:**
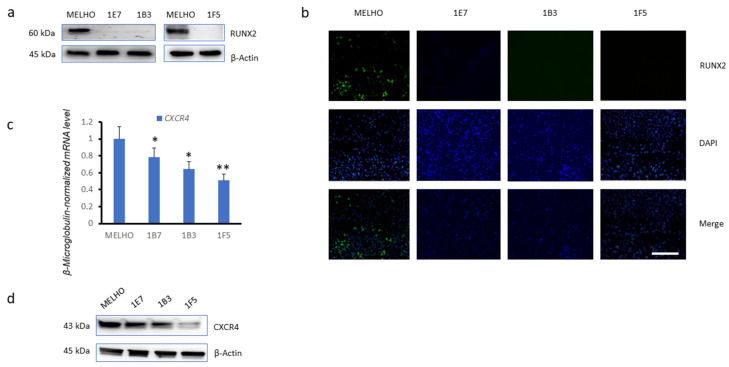
Reduced expression of *CXCR4* in *RUNX2* knockout melanoma cell lines. (**a**) Western blots of RUNX2 protein levels in the MELHO wild-type melanoma cell line and in the MELHO-derived 1E7, 1B3, and 1F5 *RUNX2* knockout (KO) cell lines. (**b**) Immunofluorescence analysis of RUNX2 protein levels in the original MELHO cell line and in the *RUNX2* KO cell lines. Scale bar represent 750 µm. (**c**) Real-time quantitative PCR and (**d**) Western blots of CXCR4 levels in the original MELHO cell line and in the MELHO-derived 1E7, 1B3, and 1F5 *RUNX2* KO cell lines. In panels a and d, β-actin was used as a loading control. mRNA expression levels were evaluated in three independent experiments. Western blot images are representative of three independent experiments. Magnification 10×. Asterisks indicate statistically significant differences (*: *p* < 0.05; **: *p* < 0.005). Original blots are presented in Appendix A.

**Figure 3 cells-13-00408-f003:**
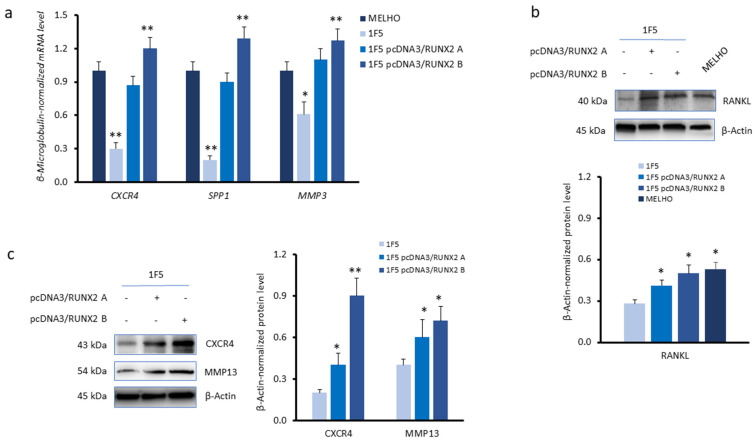
Expression of *CXCR4* and invasiveness markers in wild-type and *RUNX2* KO melanoma cell lines without or under/overexpression of RUNX2. (**a**) Real-time quantitative PCR analysis of mRNA levels of *CXCR4*, secreted phosphoprotein 1 (*SPP1*), and matrix metalloproteinase 3 (*MMP3*). Data were normalised to the β2-microglobulin housekeeping gene. (**b**,**c**) Western blots and densitometry analyses of (**b**) receptor activator of nuclear factor κΒ ligand (RANKL) and (**c**) CXCR4 and matrix metalloproteinase 13 (MMP13), proteins in the 1F5 *RUNX2* KO cell line, upon transient transfection with a RUNX2-expressing plasmid (pcDNA3/RUNX2). Two plasmid concentrations (A: 1.3 µg/mL and B: 2.6 µg/mL) were used. β-Actin was used as a loading control. mRNA levels were evaluated in three independent experiments. Western blot images are representative of three independent experiments. Asterisks indicate statistically significant differences relative to MELHO or IF5 cells (*: *p* < 0.05; **: *p* < 0.005). Original blots are presented in Appendix A.

**Figure 4 cells-13-00408-f004:**
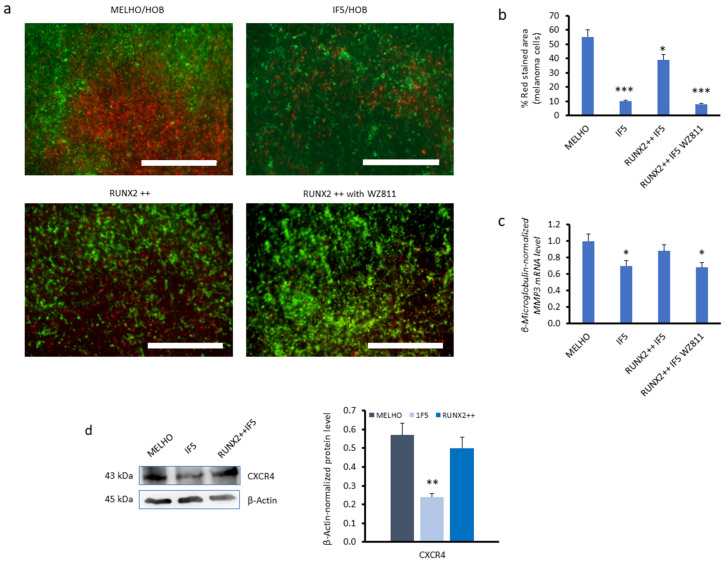
Adherence of melanoma cells to human osteoblasts (HOB) in a fluidic 3D cell culture model system. MELHO are wild-type melanoma cells, while IF5 is a *RUNX2* KO cell line, in which a *RUNX2*-expression plasmid was transfected (RUNX2 ++), in the absence or presence of the WZ822 CXCR4 inhibitor. (**a**) Representative fluorescence microscopy images and (**b**) quantification. The graph related to 3D images shows the percentage of melanoma cells that adhered to HOB (six randomly selected squares with dimensions of 100 µm^2^ were analysed using Image J software (Version 1.46r, National Institutes of Health, Bethesda, MD, USA)). Scale bars represent 1000 µm. (**c**) Real-time quantitative PCR showing the relative *MMP3* gene expression levels. (**d**) Western blots (**left**) and densitometry analyses (**right**) of CXCR4 protein in MELHO and RUNX2-restored 1F5 cells. Real-time quantitative PCR data were normalised to the β2-microglobulin housekeeping gene. mRNA levels were evaluated in three independent experiments. Western blot images are representative of three independent experiments. Asterisks indicate statistically significant differences relative to MELHO cells (*: *p* < 0.05; **: *p* < 0.005; ***: *p* < 0.001). Original blots are presented in Appendix A.

**Figure 5 cells-13-00408-f005:**
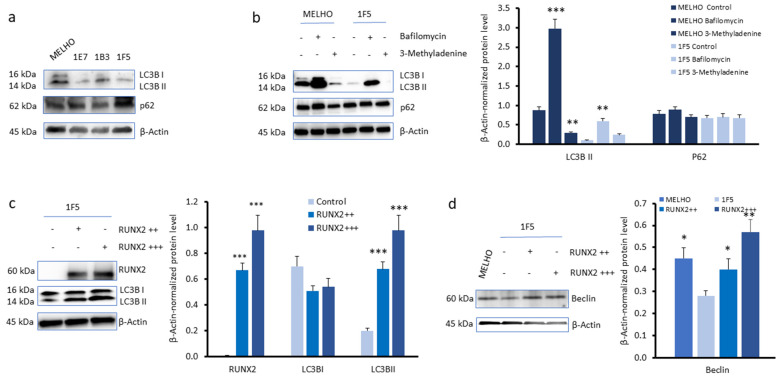
Impact of RUNX2 expression on autophagy markers in melanoma cells. (**a**) Western blots of 1A/1B-light chain 3 (LC3) and p62 in MELHO and in the 1E7, 1B3, and 1F5 *RUNX2* KO melanoma cell lines. (**b**) Western blots (**left**) and densitometry analyses (**right**) of LC3 and p62 protein levels in MELHO and in the 1F5 *RUNX2* KO cell line mock-treated or treated with bafilomycin A1 or 3-methyladenine. (**c**) Western blots (**left**) and densitometry analyses (**right**) of RUNX2 and LC3 in the 1F5 cell line, in the absence or presence of transfection with the pcDNA3/RUNX2 plasmid at 1.3 µg/mL (RUNX2 ++) or 2.6 µg/mL (RUNX2 +++). (**d**) Western blots (**left**) and densitometry analyses (**right**) of beclin in MELHO and in 1F5 cells without or with transfection with the pcDNA3/RUNX2 plasmid. β-Actin was used as a loading control. Western blot images are representative of three independent experiments. Asterisks indicate statistically significant differences relative to the control ((**b**,**c**), **right**) or 1F5 *RUNX2* KO melanoma cells (((**d**), **right**); *: *p* < 0.05; **: *p* < 0.005; ***: *p* < 0.001). Original blots are presented in Appendix A.

**Figure 6 cells-13-00408-f006:**
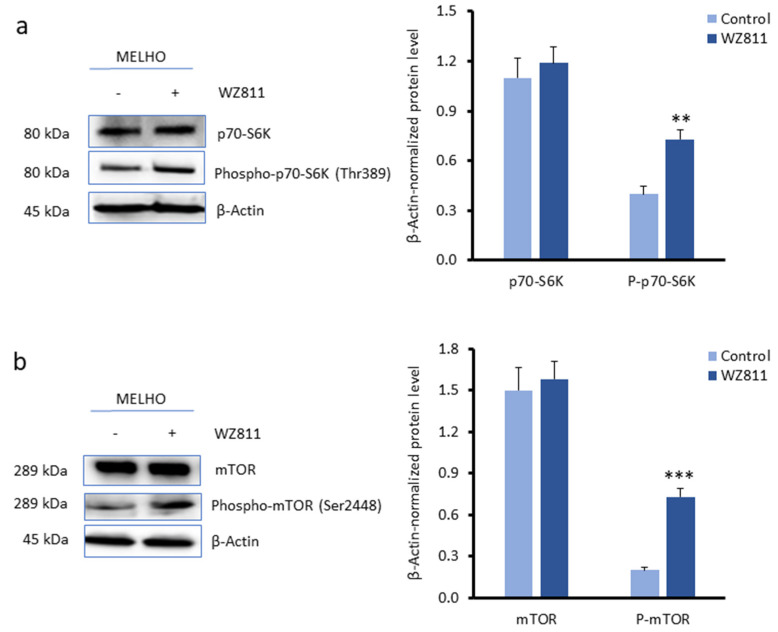
Effects of CXCR4 inhibition on the mammalian target of rapamycin (mTOR) signalling pathway. (**a**,**b**) Western blots (**left**) and densitometry analyses (**right**) of (**a**) p70-SK6 and phosphorylated p70-S6K and (**b**) mTOR and phosphorylated mTOR in MELHO cells without or under CXCR4 inhibition with WZ811. Western blot images are representative of three independent experiments. Asterisks indicate statistically significant differences relative to mock-treated cells (**: *p* < 0.005; ***: *p* < 0.001). Original blots are presented in Appendix A.

**Figure 7 cells-13-00408-f007:**
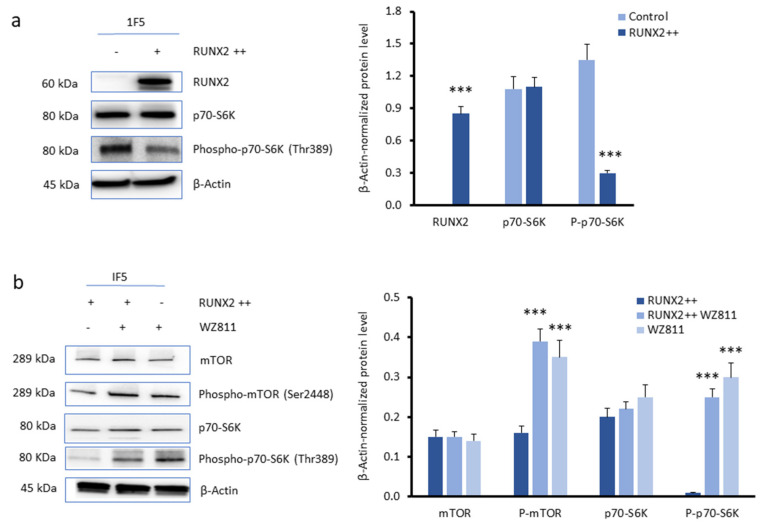
Inhibition of the mTOR signalling pathway by RUNX2 and CXCR4. (**a**,**b**) Western blots (**left**) and densitometry analyses (**right**) of (**a**) RUNX2, p70-SK6, and phosphorylated p70-S6K and of (**b**) mTOR, phosphorylated mTOR, p70-SK6, and phosphorylated p70-S6K protein levels in 1F5 cells, without or with pcDNA3/RUNX2 transfection (2.6 µg/mL) or under CXCR4 inhibition with WZ811. β-Actin was used as a loading control. Western blot images are representative of three independent experiments. Asterisks indicate statistically significant differences relative to control cells (***: *p* < 0.001). Original blots are presented in Appendix A.

**Figure 8 cells-13-00408-f008:**
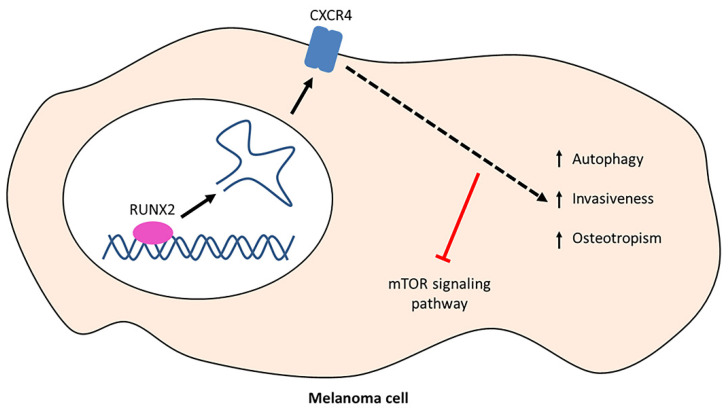
A model of RUNX2 and CXCR4 effects on melanoma cells. Melanoma cells express the RUNX2 transcription factor (represented in purple), which binds to the *CXCR4* gene (represented in blue) and promotes mRNA transcription and CXCR4 biosynthesis. The CXCR4 protein exerts multiple effects on melanoma cells, including increased autophagy, cell invasiveness, and osteotropism, with a simultaneous inhibition of the mTOR signalling pathway (represented in red).

## Data Availability

All data generated or analysed during this study are included in this published article (or in Appendix A).

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
