# Peer review of "Exploring the Interplay of RUNX2 and CXCR4 in Melanoma Progression"

_cells, 2024, doi:10.3390/cells13050408_

Round 1

Reviewer 1 Report (Previous Reviewer 1)

Comments and Suggestions for Authors

None

Author Response

Thank you  for revising our manuscript

Reviewer 2 Report (Previous Reviewer 3)

Comments and Suggestions for Authors

The raised questions have been addressed and the new version of the manuscript is sufficiently improved

Author Response

Thank you  for revising our manuscript

Reviewer 3 Report (New Reviewer)

Comments and Suggestions for Authors

The manuscript "Exploring the Interplay of RUNX2 and CXCR4 in Melanoma Progression" by Carbonare et al investigates the role of RUNX2 transcription factor to promote the expression of the CXCR4 chemokine receptor on melanoma cells, which in turn promotes autophagy, cell invasiveness, and osteotropism, through the inhibition of the mTOR signaling. The data suggest that RUNX2 promotes the melanoma progression by up-regulating CXCR4, and identify the latter as a key player in melanoma-related osteotropism.

The manuscript is well prepared, and the results are presented clearly.

Gene names should be written in italics.

Fig. 1 - why mRNA levels are inconsistent with the levels of corresponding proteins?

Fig. 2 - scale bars are missing in the microphotographs.

Discussion is written comprehensively, and the appropriate references are cited.

In general, the majority of panels in the figures, especially graphs, should be enlarged for better clarity.

Author Response

Thank you  for revising our manuscript.

Gene names should be written in italics.

Reply: As requested, I have appropriately written the gene names

Fig. 1 - why mRNA levels are inconsistent with the levels of corresponding proteins?

Reply: Multiple studies have observed a discrepancy in the quantities of matching protein and RNA molecules (Wang, Degeng. "Discrepancy between mRNA and protein abundance: insight from information retrieval process in computers." Computational biology and chemistry 32.6 (2008): 462-468.). Proteins and RNA molecules represent distinct phases in the multi-step flow of cellular genetic information. Their dynamic generation and degradation suggest that this disparity may originate from various stages in their synthesis and breakdown. Therefore, I believe that the inconsistency can be explained as a manifestation of different events in the production and degradation of these two macromolecules, reflecting the complexity of the biological processes involved and the intricate regulation that governs them.

Fig. 2 - scale bars are missing in the microphotographs

Reply: As requested, I have added the scale bars to Figure 2

In general, the majority of panels in the figures, especially graphs, should be enlarged for better clarity

Reply: As requested, I have enlarged the figures

This manuscript is a resubmission of an earlier submission. The following is a list of the peer review reports and author responses from that submission.

Round 1

Reviewer 1 Report

Comments and Suggestions for Authors

This Is an interesting and well structured study investigating

The relationship between RUNX2 and CXCR4. The article is well

Written and the conclusions are consistent.witj the experimental data.

The potential clinical application of the experimental findings

Should be Better expalined.

Reviewer 2 Report

Comments and Suggestions for Authors

In this manuscript, authors have described that RUNX2 promotes melanoma progression through upregulating CXCR4 and thereby autophagy, cell invasiveness, and osteotropism.

Comments:

1.       Minor English editing is required.

2.       Lot of typos in the manuscript text that needs to be taken care of, especially in section 3.2.

3.       Did the authors do a dose-dependent assay for the CXCR4 inhibitor? Data? On what basis 10uM conc. was decided.

4.       Protein expression for RUNX2 was higher in COLO800 so why MELHO was used as a control cell line? MELHO protein expression for RUNX2 in Fig 1d?

5.       Western blot image quality is not up to the mark and needs improvement.

6.       Was there any specific reason to pick IF5 for immunofluorescence staining? Why not 1E7 and 1B3

7.       Figure 2c needs to have RUNX2 control run at the same time with the same cell lysates. Also, 1E7 is not a good example of CXCR4 decrease.

8.       Fig 3b, why SSP-1 was not checked at the protein level?

9.       Line 280 is it methionine or 3-methyl adenine? Typo?

10. Study lacks the in vivo experiments that must have been done with the RUNX-2 KO cell lines to show how it affects melanoma growth

Comments on the Quality of English Language

Minor English editing and typo needs to be checked throughout the manuscript.

Reviewer 3 Report

Comments and Suggestions for Authors

In this manuscript, Luca Dalle Carbonare and collaborators investigated the interplay of RUNX2 and CXCR4 in melanoma. In general, the research subject is of interest. However, part of the results is not convincing because the low quality of some immunoblots and more experiments are required to support the conclusions.

     1)     In general, the protein marker indicating the molecular weight of proteins is missing and should be added in all WB.

2)     Fig 2C: the expression of CXCR4 protein analysed by WB is not convincing. Based on the original WB, it is quite difficult to establish which band correspond to the CXCR4 protein. This is an important experiment and authors should repeat it and provide a more convincing WB showing the effect of RUNX2 depletion on CXCR4 expression. In addition, did the authors test the mRNA expression of CXCR4 by RT-PCR in all indicated RUNX2-KO cells?

3)     Fig 3b: to better understand the correlation between the expression levels RUNX2 and CXCR4, the expression of all indicated proteins in MELHO wild type should be added. Is the expression of CXCR4 in 1FS rescued cells comparable with MELHO wild type cells?

4)     Fig 5C. The authors should explain why there are two bands of RUNX2.

5)     Fig 5C. The original blot of Beclin contains many bands and the data is not convincing. How the authors decided which band is the protein Beclin? The marker indicating the molecular weight is missing. This experiment should be repeated using a better anti-Beclin for immunoblot and a positive control for Beclin.

6)     English editing revision is required.

Comments on the Quality of English Language

English editing revision is required.